# Extremely Low Activity of Serum Alanine Aminotransferase Is Associated with Long-Term Overall-Cause Mortality in the Elderly Patients Undergoing Percutaneous Coronary Intervention after Acute Coronary Syndrome

**DOI:** 10.3390/medicina59020415

**Published:** 2023-02-20

**Authors:** Doo Young Kim, Si-Woon Park, Hyung-Wook Han, Myeong-Kon Kim, Ha-Jung Kim

**Affiliations:** 1The Convergence Institute of Healthcare and Medical Science, Catholic Kwandong University College of Medicine, Incheon 22711, Republic of Korea; 2Department of Rehabilitation Medicine, International St. Mary’s Hospital, Catholic Kwandong University College of Medicine, Incheon 22711, Republic of Korea; 3Department of Cardiology, International St. Mary’s Hospital, Catholic Kwandong University College of Medicine, Incheon 22711, Republic of Korea; 4Department of Anesthesiology and Pain Medicine, Asan Medical Center, University of Ulsan College of Medicine, Seoul 05505, Republic of Korea

**Keywords:** acute coronary syndrome, alanine transaminase, frailty, mortality, percutaneous coronary intervention

## Abstract

*Background and Objectives*: Recent studies revealed that the extremely low activity of serum alanine aminotransferase (ALT) is associated with frailty and contributes to increased mortality after acute physical stress. We aimed to investigate whether the extremely low activity of serum ALT (<10 U/L) at the time of diagnosis can be used to predict overall-cause mortality in elderly patients that underwent percutaneous coronary intervention (PCI) after acute coronary syndrome (ACS) diagnosis. *Materials and Methods*: A retrospective medical record review was performed on 1597 patients diagnosed with ACS who underwent PCI at a single university hospital from February 2014 to March 2020. The associations between the extremely low activity of serum ALT and mortality were assessed using a stepwise Cox regression (forward: conditional). *Results*: A total of 210 elderly patients were analyzed in this study. The number of deaths was 64 (30.5%), the mean survival time was 25.0 ± 18.9 months, and the mean age was 76.9 ± 7.6 years. The mean door-to-PCI time was 74.0 ± 20.9 min. The results of stepwise Cox regression analysis showed that the extremely low activity of serum ALT (adjusted hazard ratio: 5.157, 95% confidence interval: 3.001–8.862, *p* < 0.001) was the independent risk factor for long-term overall-cause mortality in the elderly who underwent PCI after ACS diagnosis. *Conclusions*: The extremely low activity of serum ALT at ACS diagnosis is a significant risk factor for increased long-term overall-cause mortality in the elderly who underwent PCI after ACS diagnosis. It is noteworthy that a simple laboratory test at the time of diagnosis was found to be a significant risk factor for mortality.

## 1. Introduction

In general, serum alanine aminotransferase (ALT) activity is one of the well-known routine laboratory tests, with activity levels above the reference range indicating injury to the liver [1]. Like albumin, ALT is a biomolecular substance synthesized in the liver. Thus, as the liver ages, the synthesis of albumin and ALT decreases, resulting in a decrease in serum concentrations [2]. Recent studies revealed that the extremely low activity of serum ALT and the low level of serum albumin are associated with frailty and contribute to increased mortality after acute physical stress [2,3,4,5]. A meta-analysis reported that the lower the serum ALT activity in the elderly after acute stress, the higher is the mortality rate; a significant increase in mortality for every five units decrease from the reference range level was also reported [6].

Frailty refers to a state wherein the ability to maintain physical homeostasis is decreased and stress vulnerable [7]. It is known that about one-third of the elderly, even without any disease, have frailty [8]. Although frailty itself does not increase mortality, after severe physical stress, people with frailty are vulnerable to stress, and their physical homeostasis might not be restored, leading to death [9]. Acute coronary syndrome (ACS) is one of the most severe physical stress conditions [10]. The estimated overall prevalence of ACS in the United States is approximately 3.1% among individuals aged 20 years or older, contributing to more than 100,000 fatalities annually [11]. In patients diagnosed with ACS under extreme physical stress as a clinical symptom, percutaneous coronary intervention (PCI) is the first choice of treatment [12]. The patients who underwent PCI after ACS diagnosis experienced severe physical stress even after recovering from acute illnesses; this stress may limit the recovery of physical homeostasis in patients with frailty and may lead to death [13,14].

Frailty and ACS are both common among the elderly [15]. Moreover, although frailty has a negative effect on the prognosis of ACS patients, there is no fast and reliable predictor for determining it in emergencies. Therefore, we investigated the main question of our study: whether the extremely low activity of serum ALT is useful as a predictor of long-term overall-cause mortality in elderly patients undergoing percutaneous coronary intervention after acute coronary syndrome.

## 2. Materials and Methods

### 2.1. Study Design and Sample Size Calculation

This study was a retrospective cohort study that reviewed electronic medical records generated during treatment at a single university hospital [16]. G*power 3.1 software was used for a priori power analysis to determine an appropriate sample size [17]. The parameters set were a power of 0.95 and an alpha error rate of 0.05, based on the effect size derived from the previous study [5]. The minimum sample size was calculated as 199 individuals, including 18 patients with extremely low serum ALT activity and 181 patients with normal ALT values.

### 2.2. Participants

The medical records of patients who were diagnosed with ACS and had undergone PCI from February 2014 to March 2020 were retrospectively reviewed. All PCIs were performed at a single university hospital in Incheon, South Korea. A total of 1597 patients were identified through our electronic medical record system. Of these, the following patients were excluded: patients with a previous ACS history (*n* = 279), patients under the age of 65 years (*n* = 959), patients with suspected or diagnosed liver disease (ALT > 40, *n* = 84), and patients with insufficient medical record data (*n* = 65). Finally, 210 subjects were included in the analysis (Figure 1).

### 2.3. Clinical Data

We obtained the demographic data, including age, sex, body mass index (BMI), medical history, including underlying disease (diabetes, hypertension, cerebrovascular diseases, and chronic kidney disease), and smoking history. We also collected the results of diagnostic tests and the risk factors for mortality after ACS as follows: (1) door-to-PCI time, (2) number of invading vessels, (3) initial laboratory test results at the diagnosis (ALT, total cholesterol, high-density lipoprotein [HDL], hemoglobin [Hb], glucose, creatinine, c-reactive protein [CRP], N-terminal pro-brain natriuretic peptide [NT-ProBNP], troponin I high sensitivity [TnI], and creatine kinase-MB [CK-MB]), (4) initial systolic blood pressure (SBP) and electrocardiogram findings of ST elevation, and (5) left ventricular hypertrophy (LVH) on echocardiogram. A mortality survey was conducted using the National Health Insurance database as of 30 October 2020.

### 2.4. Statistical Analyses

We compared the baseline characteristics of the death group and the surviving group. Mann–Whitney U-test or independent *t*-test was used for continuous variables, and Fisher’s exact test or chi-square test was used for categorical variables. The continuous variable was described as the mean standard deviation of the variable with normal distribution and as the median value (1st quartile, 3rd quartile) of the variable without normal distribution. The categorical variables were described as frequency and percentage. Kaplan–Meier survival curve and log-rank test were conducted, including factor analysis for assessing the activity of serum ALT.

The associations between the extremely low activity of serum ALT and mortality were assessed using a stepwise Cox regression (forward: conditional). The following covariates were included in the model to adjust confounding factors: age, sex, BMI, door-to-PCI time, number of invading vessels, LVH on echocardiogram, smoking history, underlying disease (diabetes, hypertension, chronic kidney disease, and cerebrovascular disease), and initial laboratory test results at the diagnosis as follows: low levels of Hb group (<11 g/dL), low levels of HDL group (<40 mg/dL), total cholesterol (mg/dL), creatinine (mg/dL), CRP (mg/L), glucose (mg/dL), CK-MB (pg/mL), NT-Pro-BNP (pg/mL), and TnI high sensitivity (pg/mL). Additionally, SBP and electrocardiogram findings of ST elevation at the time of diagnosis were obtained. To test the assumption in the Cox proportional hazard model, visual inspection of the curve and time-dependent Cox regression were performed. The two-sided *p*-values less than 0.05 were considered statistically significant. All statistical analyses were performed using the SPSS version 22.0 (IBM Corp., Armonk, NY, USA).

## 3. Results

### 3.1. Clinical Description

A total of 210 elderly patients were analyzed in this study. The number of patients who died during the study period was 64 (30.5%). The mean survival time was 25.0 ± 18.9 months, and the mean age was 76.9 ± 7.6 years. The mean door-to-PCI time was 74.0 ± 20.9 min. The variables with significant differences between the death and surviving groups were age, BMI, survival time, Hb, CRP, HDL, NT-ProBNP, TnI, SBP at the time of diagnosis, and the extremely low activity of serum ALT (Table 1).

### 3.2. Survival Analyses

The Kaplan–Meier survival curve revealed that mortality differed according to the activity of serum ALT (Figure 2). The difference in mortality was higher in the extremely low activity of the serum ALT group (<10 U/L) than in the normal group (10–40 U/L), which was significant immediately after onset and increased with time. The one-year survival rate of the normal group (10–40 U/L) was 84.4%, and the mean survival time was 53.4 months; the one-year survival rate of the extremely low activity of the serum ALT group (<10 U/L) was 36.4%, and the mean survival time was 15.2 months. The log-rank test result was significant (*p* < 0.001).

Unadjusted Cox regression analysis showed that age, extremely low activity of serum ALT, and NT-ProBNP were associated with all-cause mortality after PCI (Table 2). The results of the adjusted Cox proportional hazards analysis (forward: conditional), including all covariates used in the unadjusted Cox analysis, are presented in Table 3. The extremely low activity of serum ALT was a significant risk factor for long-term all-cause mortality in the elderly who underwent PCI after ACS diagnosis, even with statistical adjustments for other confounding factors (adjusted hazard ratio: 5.157, 95% Confidence Interval: 3.001–8.862, *p* < 0.001). The serum CRP and NT-ProBNP levels were also significant factors for mortality (Table 3).

## 4. Discussion

We aimed to elucidate that the extremely low activity of serum ALT could predict long-term overall-cause mortality as a surrogate marker of frailty. The results of this study revealed that CRP, NT-ProBNP, and the extremely low activity of serum ALT were independent risk factors for long-term overall-cause mortality in the elderly who had undergone PCI after ACS diagnosis. It is noteworthy that a simple laboratory test at the time of diagnosis was found to be a significant risk factor for mortality despite statistical adjustment for known risk factors.

The mortality rate in patients with ACS has decreased with improved management in the acute phase. However, ACS is still the leading cause of mortality in elderly patients and is a rather stressful event [13]. According to a 2001 study by Pocock et al., risk factors for mortality in patients with cardiovascular disease were age, smoking, SBP, total cholesterol, diabetes, Cr, LVH, and cardiac enzymes [18]. As per the result of our study, age, extremely low activity of serum ALT, and NT-ProBNP appeared to be risk factors for overall-cause mortality in the unadjusted model. However, in a model controlled for confounders using stepwise regression, significant risk factors for long-term mortality in patients undergoing PCI after ACS diagnosis were a cardiac enzyme, CRP, and extremely low activity of serum ALT. These results indicate that the severity of the myocardial injury, the systemic inflammatory response, and the extremely low activity of serum ALT have a greater effect on mortality than chronologically old age in patients undergoing PCI after ACS diagnosis.

Age is known to be associated with poor outcomes in various clinical settings for the elderly. However, chronological age could not precisely reflect each individual’s aging process, which was influenced by one’s genetic makeup and environmental factors [19]. Thus, there has been a growing interest in the biological age, which is more relevant to frailty [20]. There have been several studies that evaluated the impact of frailty on adverse outcomes in patients with ACS [21,22]. Most of the studies showed that frail patients had a higher risk for mortality and prolonged hospital stay after ACS [21,22]. This is partially because frail patients were less likely to undergo coronary angiography and PCI after ACS diagnosis compared with non-frail patients [23]. The frail patients were relatively older and had various comorbidities with multiple medications; this could have led them to refuse or apprehend invasive procedures [14]. However, a large body of evidence proved that aggressive treatment improved clinical outcomes following ACS, even in frail patients [24]. Thus, nowadays, PCI is strongly recommended for these patients [25].

However, the patients with frailty should be managed more carefully following PCI; these patients were at high risk of postprocedural morbidities and mortality. For frail patients, there should be acute management related to bleeding issues immediately after the procedure, [26] and it is also necessary to provide a tailored rehabilitation program for better long-term prognosis post-acute rehabilitation [27,28]. To identify frailty, various screening tools have been used, including the FRAIL Scale, Frailty Index, Clinical Frailty Scale, Groningen Frailty Indicator, Vulnerable Elders Survey-13, and Edmonton Frailty Scale [29]. Most of the screening tools are based on self-reported questionnaires and/or physical performance tests such as muscle power and gait speed [29]. However, the ACS patients were sometimes unconscious or had cognitive dysfunction. Additionally, in many cases, ACS patients were incapable of undergoing some tests, including the evaluation of muscle power and gait speed. Therefore, a simple objective screening tool is required, especially if the value could be obtained through routine laboratory tests.

Among various laboratory findings, ALT was shown to be associated with frailty in several previous studies [2,30]. Hepatic organs degenerate in frail older people, reducing ALT synthesis in the liver and consequently significantly reducing serum concentrations [2]. The extremely low activity of serum ALT is associated with frailty as observed in previous studies and is an independent risk factor for overall-cause mortality in the elderly [2,4,5,6,30]. A meta-analysis showed that for every 5 U/L decrease in the activity of serum ALT, there was a significant increase in mortality among the elderly [6]. In this study, the extremely low activity of serum ALT was defined as ALT < 10 U/L, and as in the previous study, it had a significant association with mortality. A previous study demonstrated that low ALT was associated with various morbidity and mortality in ACS patients admitted to intensive cardiac care units [31]. It was consistent with our study results. However, the previous study included patients with or without coronary intervention, while our study only included patients who required percutaneous coronary intervention.

Cardiac rehabilitation after ACS is a well-received concept with proven improvements in long-term prognosis, including mortality [27]. However, the prognosis for frail patients is three times worse than that for non-frail patients; thus, frailty needs to be evaluated for older patients before cardiac surgery or procedures, even if it is time consuming [32,33]. Therefore, the long-term rehabilitation plan after cardiac surgery or intervention for frail patients should be well designed according to the patient’s condition; even pre-habilitation, the concept of preoperative rehabilitation is a recent trend [34,35]. Pre-habilitation before heart surgery or intervention has the effect of reducing the length of hospital stay by preventing the decline of physical function. It is recommended to increase the capacity of physical function through exercise during the waiting period for elective procedures [34]. However, in the case of an emergency such as ACS (not a planned intervention), the door-to-PCI time also significantly affects the prognosis; thus, a time-consuming preoperative frailty evaluation is almost impossible. The strength of this study is that we found a reliable objective indicator to estimate the frailty of ACS patients with only basic laboratory tests, so that it can be used in emergency situations instead of time-consuming tests. Physicians who performed PCI in patients after ACS should presume that patients with extremely low activity of serum ALT are a high-risk group for frailty; they should be referred to the Department of Rehabilitation Medicine immediately after the procedure for an intensive rehabilitation approach. In older, frail patients, several physiological benefits, including an increase in exercise capacity and lean mass, can be expected from rehabilitation exercise (as a therapeutic strategy) after heart disease [28].

This study has some limitations. First, the cause of death was not identified; thus, it is difficult to prove whether the death was due to ACS or frailty. However, as described above, frailty itself does not cause death but implies an overall condition that can lead to death due to insufficient recovery from acute stress; thus, we investigated the overall effect of frailty on the long-term prognosis of ACS. Second, due to the small sample size, subgroup analysis according to the characteristics of ACS could not be performed. Although the results of this study did not elucidate a clear mechanism, they will help design a prospective study by presenting persuasive data related to the prognosis of ACS, thereby improving the algorithm related to the long-term prognosis of ACS in the future.

## 5. Conclusions

The extremely low activity of serum ALT at ACS diagnosis is a significant risk factor for increased long-term, overall-cause mortality in the elderly who underwent PCI after ACS diagnosis. It is noteworthy that a simple laboratory test at the time of diagnosis was found to be a significant risk factor for mortality.

## Figures and Tables

**Figure 1 medicina-59-00415-f001:**
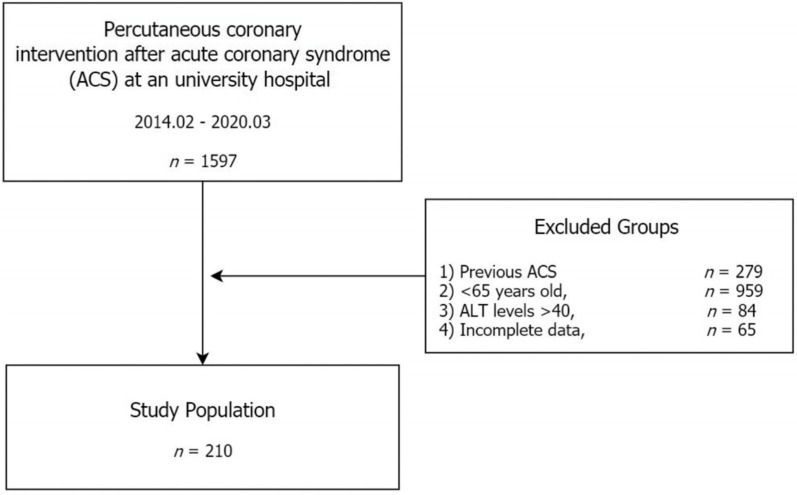
Flow diagram of this study.

**Figure 2 medicina-59-00415-f002:**
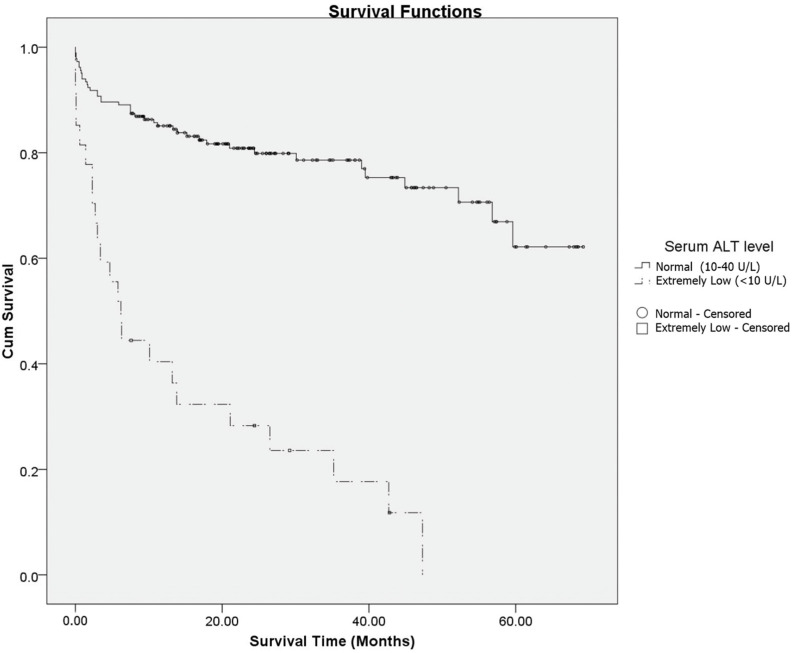
The Kaplan–Meier curves for overall survival according to the activity of serum ALT (log-rank test; *p* < 0.001).

**Table 1 medicina-59-00415-t001:** General characteristics and comparison between death and surviving groups.

	Total (*n* = 210)	Surviving (*n* = 146)	Death (*n* = 64)	*p*-Value
Age (years)	76.9 ± 7.6	75.2 ± 7.0	80.7 ± 7.4	<0.001 *
Survival Time (month)	25.0 ± 18.9	30.6 ± 17.5	12.3 ± 15.9	<0.001 *
Sex, *n* (%)				0.522
-Female	98 (46.7%)	66 (45.2%)	32 (50.0%)	
-Male	112 (53.3%)	80 (54.8%)	32 (50.0%)	
BMI, *n* (%)				0.002 *
-Reference (18.5–24.9)	127 (60.5%)	85 (58.3%)	42 (65.6%)	
-Underweight (<18.5)	15 (7.1%)	5 (3.4%)	10 (15.6%)	
-Overweight (25.0–29.9)	62 (29.4)	51 (34.9%)	11 (17.2%)	
-Obese (≥30)	6 (2.9%)	5 (3.4%)	1 (1.6%)	
Smoking history, *n* (%)				0.262
-Non-smoker	144 (68.6%)	104 (71.2%)	40 (62.5%)	
-Past-smoker	41 (19.5%)	28 (19.2%)	13 (20.3%)	
-Current-smoker	25 (11.9%)	14 (9.6%)	11 (17.2%)	
Invading vessel, *n* (%)				0.639
-Single vessel	165 (78.6%)	116 (79.5%)	49 (76.6%)	
-Multiple vessels	45 (21.4%	30 (20.5%)	15 (23.4%)	
Door-to-PCI time (minutes)	74.0 ± 20.9	72.2 ± 16.6	77.9 ± 28.0	0.133
ST elevation on ECG, *n* (%)	62 (29.5%)	39 (26.7%)	23 (35.9%)	0.177
LVH on echocardiogram, *n* (%)	44 (21.0%)	26 (17.8%)	18 (28.1%)	0.091
Initial SBP at diagnosis (mmHg)	131.2 ± 27.2	134.3 ± 25.9	124.1 ± 28.7	0.012 *
Laboratory findings at diagnosis				
-Extremely low activity of serum ALT (<10 U/L), *n* (%)	27 (12.9%)	4 (2.7%)	23 (35.9%)	<0.001*
-ALT (U/L)	19.7 ± 8.5	20.14 ± 7.8	18.8 ± 9.9	0.358
-Creatinine (mg/dL)	1.2 ± 1.0	1.1 ± 0.9	1.4 ± 1.2	0.089
-CRP (mg/L)	23.2 ± 46.9	11.1 ± 27.4	50.6 ± 66.9	<0.001 *
-Hb (g/dL)	12.7 ± 2.1	13.1 ± 1.9	12.0 ± 2.3	<0.001 *
-Glucose (mg/dL)	174.9 ± 74.6	171.0 ± 73.1	183.7 ±77.6	0.256
-Total cholesterol (mg/dL)	161.5 ± 43.7	162.7 ± 40.8	158.9 ± 49.9	0.569
-HDL (mg/dL)	42.4 ± 9.7	43.3 ± 9.4	40.19 ± 10.2	0.030 *
-CK-MB (pg/mL)	5.7 ± 10.8	4.7 ± 10.5	8.0 ± 11.2	0.043 *
-NT-ProBNP (pg/mL)	886.2 [176.6, 3777]	444.3 [119.4, 1743]	3500.5 [983.6, 12766]	<0.001 *
-TnI (pg/mL)	47.9 [9.8, 675.2]	28.5 [7.9, 213]	525.4 [25.0, 2219.2]	<0.001 *
Underlying diseases				
-DM, *n* (%)	90 (42.9%)	64 (43.8%)	26 (40.6%)	0.665
-Hypertension, *n* (%)	145 (69.1%)	100 (68.5%)	45 (70.3%)	0.793
-Chronic kidney disease, *n* (%)	15 (7.1%)	9 (6.2%)	6 (9.4%)	0.397
-Cerebrovascular disease, *n* (%)	18 (8.6%)	11 (7.5%)	7 (10.9%)	0.417

Values, mean ± standard deviation or median [1st quartile, 3rd quartile]; ALT, alanine aminotransferase; BMI, body mass index; PCI, percutaneous coronary intervention; ECG, electrocardiogram; LVH, left ventricular hypertrophy; SBP, systolic blood pressure; CRP, c-reactive protein; Hb, hemoglobin; HDL, high-density lipoprotein; CK-MB, creatine kinase-MB; NT-proBNP, N-terminal pro-brain natriuretic peptide; TnI, troponin I high sensitivity; DM, diabetes mellitus; * *p* < 0.05.

**Table 2 medicina-59-00415-t002:** The Results of unadjusted Cox regression analysis for predicting mortality in patients with ACS.

	Hazard Ratio	95% Confidence Interval		*p*-Value
		Lower	Upper	
Age (years)	1.062	1.016	1.110	0.008 *
Sex (reference; female), *n*	1.109	0.565	2.180	0.763
Extremely low activity of serum ALT group (<10 U/L), *n*	5.024	2.588	9.753	<0.001 *
BMI (reference; 18.5–24.9), *n*				0.635
-Underweight (<18.5)	1.408	0.611	3.245	0.421
-Overweight (25.0–29.9)	0.716	0.337	1.520	0.384
-Obese (≥30)	0.679	0.082	5.599	0.719
Smoking history (reference; non-smoker), *n*				0.453
-Past-smoker	1.293	0.552	3.031	0.554
-Current-smoker	2.432	0.594	9.964	0.217
Invading multiple vessels, *n*	0.948	0.485	1.852	0.875
Door-to-PCI time (minutes)	1.004	0.991	1.016	0.569
ST elevation on ECG, *n*	1.163	0.603	2.243	0.652
LVH on echocardiogram, *n*	1.287	0.678	2.445	0.441
Initial SBP at diagnosis (mmHg)	0.991	0.979	1.003	0.158
Laboratory findings at diagnosis				
-Low level Hb group (<11 g/dL)	1.093	0.570	2.098	0.789
-Low level HDL group (<40 mg/dL)	1.111	0.584	2.114	0.749
-Total cholesterol (mg/dL)	1.007	0.999	1.014	0.078
-Creatinine (mg/dL)	0.904	0.665	1.229	0.519
-CRP (mg/L)	1.002	0.996	1.008	0.561
-Glucose (mg/dL)	0.999	0.995	1.003	0.504
-CK-MB (pg/mL)	1.010	0.984	1.036	0.454
-NT-ProBNP (pg/mL)	1.000	1.000	1.000	0.021 *
-TnI (pg/mL)	1.000	1.000	1.000	0.680
Underlying diseases				
-DM, *n*	1.225	0.590	2.541	0.586
-Hypertension, *n*	1.457	0.735	2.890	0.281
-Chronic kidney disease, *n*	2.864	0.848	9.674	0.090
-Cerebrovascular disease, *n*	1.851	0.708	4.841	0.209

ACS, acute coronary syndrome; ALT, alanine aminotransferase; BMI, body mass index; PCI, percutaneous coronary intervention; ECG, electrocardiogram; LVH, left ventricular hypertrophy; SBP, systolic blood pressure; Hb, hemoglobin; HDL, high-density lipoprotein; CRP, c-reactive protein; CK-MB, creatine kinase-MB; NT-proBNP, N-terminal pro-brain natriuretic peptide; TnI, troponin I high sensitivity; DM, diabetes mellitus; * *p* < 0.05.

**Table 3 medicina-59-00415-t003:** The results of adjusted Cox regression analysis (forward stepwise: conditional) for predicting mortality in patients with ACS.

	Adjusted Hazard Ratio	95% Confidence Interval		*p*-Value
		Lower	Upper	
Extremely low activity of serum ALT group (<10 U/L), *n*	5.157	3.001	8.862	<0.001 *
CRP (mg/L)	1.006	1.001	1.011	0.010 *
NT-ProBNP (pg/mL)	1.000	1.000	1.000	0.003 *

ACS, acute coronary syndrome; ALT, alanine aminotransferase; CRP, C-reactive protein; NT-proBNP, N-terminal pro-brain natriuretic peptide; * *p* < 0.05. Note: By the stepwise Cox regression analysis, all confounders that were assessed are listed below, and only listed confounders were selected by stepwise selection. Adjusted confounders: age, sex, body mass index, door-to-percutaneous coronary intervention time, number of invading vessels, left ventricular hypertrophy on echocardiogram, smoking history, underlying disease (diabetes, hypertension, chronic kidney disease, and cerebrovascular disease), the initial laboratory test results at the diagnosis (extremely low activity of serum ALT group (<10 U/L), low level of hemoglobin group (<11 g/dL), low level of high-density lipoprotein group (<40 mg/dL), total cholesterol (mg/dL), creatinine (mg/dL), CRP (mg/L), glucose (mg/dL), creatine kinase-MB (pg/mL) NT-Pro-BNP (pg/mL), and troponin I high sensitivity (pg/mL)), and the systolic blood pressure and electrocardiogram findings of ST elevation at the time of diagnosis.

## Data Availability

The data that support the findings of this study are available from the corresponding author upon reasonable request.

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
