# Peer review of "Extremely Low Activity of Serum Alanine Aminotransferase Is Associated with Long-Term Overall-Cause Mortality in the Elderly Patients Undergoing Percutaneous Coronary Intervention after Acute Coronary Syndrome"

_medicina, 2023, doi:10.3390/medicina59020415_

Round 1

Reviewer 1 Report

It is my honor & pleasure reviewing this scientific paper. A well-developed scientific article that focuses on an important topic.

I would recommend using MeSH (Medical Subject Headings) to select your Keywords. Try to visit https://www.ncbi.nlm.nih.gov/mesh/

I would recommend highlighting incidence, mortality, and morbidity rate of acute coronary syndrome (ACS) both globally & locally to show the importance of your research topic.

I would recommend ending the last paragraph of your introduction with your research paper main question instead of restating your objectives.

In the method section, you should explain more to your reader about the used research method (A retrospective medical record review) try to support your choice by citing research methods articles or books.

Determining minimum sample size procedure has not been highlighted in this article.

DOI (Digital object identifier) should be included for each cited article at your reference list

Thank you very much!

Author Response

Reviewer 1

We are very grateful for your valuable comments and suggestions. We have revised our manuscript according to your recommendations.

#1. I would recommend using MeSH (Medical Subject Headings) to select your Keywords. Try to visit https://www.ncbi.nlm.nih.gov/mesh/

  • As suggested by the reviewer, we reviewed using the MeSH term in Pubmed, and confirmed that all of the previously written Keywords in our article were MeSH terms.

#2. I would recommend highlighting incidence, mortality, and morbidity rate of acute coronary syndrome (ACS) both globally & locally to show the importance of your research topic.

  • As suggested by the reviewer, the following was added to the second paragraph of the introduction.

The estimated overall prevalence of acute coronary syndrome (ACS) in the United States is approximately 3.1% among individuals aged 20 years or older, contributing to more than 100,000 fatalities annually [11].

  1. Tsao C.W., Aday A.W., Almarzooq Z.I., Alonso A., Beaton A.Z., Bittencourt M.S., Boehme A.K., Buxton A.E., Carson A.P., Commodore-Mensah Y. Heart disease and stroke statistics—2022 update: a report from the American Heart Association. Cir-culation. 2022;145:e153-e639. 10.1161/CIR.0000000000001052

#3. I would recommend ending the last paragraph of your introduction with your research paper main question instead of restating your objectives.

  • As suggested by the reviewer, we have made the following changes to the text:

Therefore, we investigated the main question of our study, whether the extremely low activity of serum ALT is useful as a predictor of long- term overall-cause mortality in elderly patients undergoing percutaneous coronary intervention after acute coronary syn-drome.

#4. In the method section, you should explain more to your reader about the used research method (A retrospective medical record review) try to support your choice by citing research methods articles or books.

  • As suggested by the reviewer, we added the following sentence to the Methods section and cited relevant reference.

"This study was a retrospective cohort study that reviewed electronic medical records generated during treatment at a single university hospital [16]. "

  1. Worster A., Haines T. Advanced statistics: understanding medical record review (MRR) studies. Acad Emerg Med. 2004;11:187-92. 10.1111/j.1553-2712.2004.tb01433.x

#5. Determining minimum sample size procedure has not been highlighted in this article.

  • We calculated the minimum sample size using G-power added the following to the text.

G*power 3.1 software was used for a priori power analysis to determine an appropriate sample size [17]. The parameters set were a power of 0.95 and an alpha error rate of 0.05, based on the effect size derived from the previous study [5]. The minimum sample size was calculated as 199 individuals, including 18 patients with extremely low serum ALT activity and 181 patients with normal ALT values.

  1. Faul F., Erdfelder E., Lang A.-G., Buchner A. G* Power 3: A flexible statistical power analysis program for the social, behavioral, and biomedical sciences. Behav Res Methods. 2007;39:175-91. 10.3758/bf03193146
  2. An S.J., Yang Y.-J., Jeon N.-m., Hong Y.-P., Kim Y.I., Kim D.-Y. Significantly reduced alanine aminotransferase level increases all-cause mortality rate in the elderly after ischemic stroke. Int J Environ Res Public Health. 2021;18:4915. 10.3390/ijerph18094915

#6. DOI (Digital object identifier) should be included for each cited article at your reference list

  • As suggested, DOI was additionally included in all possible references.

Thank you very much!

Reviewer 2 Report

Reviews to the manuscript entitled: “Extremely low activity of serum alanine aminotransferase is associated with long-term mortality in the elderly patients undergoing percutaneous coronary intervention after acute coronary syndrome”, are as follows:

1. It is not clear that what exactly caused mortality in patients. All of 210 patients in this study were old. How did the researchers diagnose that extremely low ALT caused their death?  It is possible that participants’ old age was the main reason for their death.

2. In my opinion, it was better if the researchers performed this study as a comparative cross-sectional study in which they compare patients with or without low ALT. I think this may recognize the role of low ALT more specifically    

3. It is recommended to use recent articles in the study.

4. Many other variables can increase mortality in patients with ACS. How did the researchers control these confounding variables? Please explain it.

In general, as this manuscript’s method and the main reason for patients’ death, is not clear, from my point of view, it is not appropriate for publication in the journal of Medicina and I will reconsider after major revision.

Best of luck!

Author Response

Reviewer 2

We are very grateful for your valuable comments and suggestions. We have revised our manuscript according to your recommendations.

# 1. It is not clear that what exactly caused mortality in patients. All of 210 patients in this study were old. How did the researchers diagnose that extremely low ALT caused their death?  It is possible that participants’ old age was the main reason for their death.

  • Frailty is not a direct cause of death, but rather a decline in physical function that can lead to death as a result of decreased ability to recover from illnesses or stressors. Several studies have demonstrated that frailty is associated with an increased risk of mortality, even after controlling for age, comorbidities, and other factors. Additionally, several longitudinal studies have shown that frailty is a strong predictor of adverse events. This evidence supports the notion that frailty is a declined state in physical function that can lead to decreased ability to recover from stressors and illness, ultimately increasing the risk of death. Based on this concept, in this study, we investigated the association between all-cause mortality after PCI and the extremely low ALT as an index of frailty. This concept has been described between lines 48 and 60 of the introduction section of the manuscript.

  • As the reviewer pointed out, since frailty is not a direct cause of death, it is appropriate to use the term “overall-cause mortality”. The “overall-cause mortality” was stated in the abstract and discussion, but not in the title, abstract, introduction, and conclusion. To reduce confusion, we made it clear by emphasizing “overall-cause mortality” in the title, abstract, introduction, and conclusion.

#2. In my opinion, it was better if the researchers performed this study as a comparative cross-sectional study in which they compare patients with or without low ALT. I think this may recognize the role of low ALT more specifically    

  • Survival analysis study has several advantages over cross-sectional studies when studying the relationship between frailty and health outcomes, including death. Some of these advantages are:

1) Temporal sequence: Survival analysis studies allow for the examination of events over time, which can provide a clearer understanding of the causality between frailty and health outcomes.

2) Dynamic assessments: Survival analysis studies enable the collection of repeated measures over time, which can provide a more detailed picture of changes in health status and risk factors over time.

3) Ability to control for confounding variables: Survival analysis studies allow for the control of confounding variables factors that may influence both frailty and health outcomes (Not the Kaplan–Meier, but Cox proportional regression analysis, Forward stepwise. Shown in Table 3.)

4) Analysis of time-to-event data: Survival analysis is specifically designed for the analysis of time-to-event data, making it a suitable method for examining the relationship between frailty and the time to adverse health events, including death.

  • Therefore, we can present the results of the following cross-sectional comparative analysis in a supplement, but we have yet to include them in this revision because similar results have already been presented by survival analysis.

Extremely low ALT group

Normal Group

p-value

(N=27)

(N=183)

Age (years)

81.81 ± 6.36

76.17 ± 7.48

<0.001*

Survival Time (month)

13.07 ± 14.96

26.75 ± 18.90

<0.001*

Sex, n (%)

0.432

  - F         

15 (55.56%)

83 (45.36%)

  - M         

12 (44.44%)

100 (54.64%)

Death, n (%)

23 (85.19%)

41 (22.40%)

<0.001*

BMI, n (%)

0.05

  - Reference (18.5–24.9)

17 (62.96%)

110 (60.11%)

  - Underweight (<18.5)

5 (18.52%)

10 (5.46%)

  - Overweight (25.0–29.9)

5 (18.52%)

57 (31.15%)

  - Obese (≥30)

0 ( 0.0%)

6 (3.28%)

Smoking history, n (%)

0.929

- Non-smoker

18 (66.67%)

126 (68.85%)

  - Current-smoker

3 (11.11%)

22 (12.02%)

  - Past-smoker

6 (22.22%)

35 (19.13%)

Invading vessel, n (%)

0.389

  - Single vessel

19 (70.37%)

146 (79.78%)

  - Multiple vessels

8 (29.63%)

37 (20.22%)

Door-to-PCI time (minutes)

74.59 ± 23.15

73.87 ± 20.55

0.868

LVH on echocardiogram, n (%)

5 (18.52%)

39 (21.31%)

0.937

Initial SBP at diagnosis (mmHg)

130.52 ± 27.68

131.32 ± 27.15

0.886

ST elevation on ECG, n (%)

7 (25.93%)

55 (30.05%)

0.831

Underlying diseases

- Hypertension, n (%)

22 (81.48%)

123 (67.21%)

0.203

- DM, n (%)

10 (37.04%)

80 (43.72%)

0.655

- Chronic kidney disease, n (%)

1 (3.70%)

14 (7.65%)

0.732

- Cerebrovascular disease, n (%)

4 (14.81%)

14 (7.65%)

0.383

#3. It is recommended to use recent articles in the study.

  • "The recent articles were cited and added to the sample size calculation process or manuscript."
  1. Kim D.Y., Cho K.-C. Extremely low serum alanine transaminase level is associated with all-cause mortality in the elderly after intracranial hemorrhage. J Korean Neurosurg Soc. 2021;64:460-8. 10.3340/jkns.2020.0212
  2. An S.J., Yang Y.-J., Jeon N.-m., Hong Y.-P., Kim Y.I., Kim D.-Y. Significantly reduced alanine aminotransferase level increases all-cause mortality rate in the elderly after ischemic stroke. Int J Environ Res Public Health. 2021;18:4915. 10.3390/ijerph18094915
  3. Tsao C.W., Aday A.W., Almarzooq Z.I., Alonso A., Beaton A.Z., Bittencourt M.S., Boehme A.K., Buxton A.E., Carson A.P., Commodore-Mensah Y. Heart disease and stroke statistics—2022 update: a report from the American Heart Association. Cir-culation. 2022;145:e153-e639. 10.1161/CIR.0000000000001052

  1. Many other variables can increase mortality in patients with ACS. How did the researchers control these confounding variables? Please explain it.

  • We described many variables that can increase mortality in patients with ACS and how we corrected for these confounders using statistical techniques in lines 184-194.
  • The authors used Cox regression with the Forward stepwise: Conditional method, which operates similarly to the stepwise method in linear regression analysis and automatically includes statistically significant factors. The correction of confounders is also explained in lines 113 to 122 and in Table 3.

Thank you very much!
